# Cataract-associated P23T γD-crystallin retains a native-like fold in amorphous-looking aggregates formed at physiological pH

Jennifer C. Boatz[1,2], Matthew J. Whitley[1], Mingyue Li[1,2], Angela M. Gronenborn[1,2] & Patrick C.A. van der Wel[1,2]

Cataracts cause vision loss through the large-scale aggregation of eye lens proteins as a result of ageing or congenital mutations. The development of new treatments is hindered by uncertainty about the nature of the aggregates and their mechanism of formation. We describe the structure and morphology of aggregates formed by the P23T human γD-crystallin mutant associated with congenital cataracts. At physiological pH, the protein forms aggregates that look amorphous and disordered by electron microscopy, reminiscent of the reported formation of amorphous deposits by other crystallin mutants. Surprisingly, solid-state NMR reveals that these amorphous deposits have a high degree of structural homogeneity at the atomic level and that the aggregated protein retains a native-like conformation, with no evidence for large-scale misfolding. Non-physiological destabilizing conditions used in many *in vitro* aggregation studies are shown to yield qualitatively different, highly misfolded amyloid-like fibrils.

[1] Department of Structural Biology, University of Pittsburgh School of Medicine, 3501 Fifth Avenue, Pittsburgh, Pennsylvania 15213, USA. [2] The Center for Protein Conformational Diseases, University of Pittsburgh, Pittsburgh, Pennsylvania 15213, USA. Correspondence and requests for materials should be addressed to P.C.A.v.d.W. (email: vanderwel@pitt.edu).

Cataract formation is the leading cause of blindness, and age-related and congenital cataracts affect close to 100 million people across the globe[1]. The underlying cause of eye lens opacification is the formation of protein aggregates[2]. As such it is but one example of a broad family of protein-aggregation-related diseases, in which normally soluble proteins end up as insoluble protein deposits in affected organs[3]. Current treatments of protein aggregation diseases (including cataracts) are limited to the invasive removal of affected tissue or treatment of symptoms, rather than a reversal, prevention or inhibition of the underlying aggregation process. A limiting factor in efforts to achieve the latter is our poor understanding of the molecular mechanism by which the protein aggregation occurs.

The α-, β- and γ-crystallins, which constitute 90% of the total eye lens protein mass, are found in cataract-associated protein deposits. The α-crystallins are small heat shock proteins that perform protein quality control and counteract protein aggregation, misfolding and cataract formation[4,5]. The β- and γ-crystallins are thought to be responsible for generating and maintaining the proper refractive index of the eye lens[6]. To do so, the β/γ-crystallins are present at concentrations of 400 mg ml$^{-1}$ in healthy lenses. Cataract formation occurs when the high solubility of β/γ-crystallins is compromised as a result of changes induced by mutations or chemical damage that accumulates with ageing[1,2]. However, it has proved difficult to relate specific mutations and structural changes to a molecular mechanism of cataract-associated protein aggregation[7–13]. X-ray crystallographic and solution NMR studies have yielded atomic structures for many of the structurally related β/γ-crystallins, including both wild-type (WT) proteins and proteins with either disease-associated mutations or mutations mimicking ageing-induced damage[14–18]. Figure 1 shows the structure of the P23T mutant of human γD-crystallin (hγD), with the Greek key motifs that characterize the native fold of the γ-crystallins[17,18]. As examined in detail in prior work[12,17–20], this mutant protein is structurally very similar to the WT protein. This is reflected in a backbone root mean squared deviation of 0.4 and 0.5 Å for the N-terminal (NTD; cyan) and C-terminal domains (CTD; pink), respectively (comparing the X-ray structures), or solution NMR per-residue chemical shift differences (Δδ) of <0.2 p.p.m. (except for the 0.4 p.p.m. Δδ for the residue after the P23T mutation)[17,18]. Nonetheless, the mutant protein has lost the remarkable solubility of WT hγD and is associated with autosomal dominant congenital cataracts that form early in a child's development, affecting families across the globe[12,17–21].

Based on in vitro studies, two prominent mechanisms have been put forward to explain the aggregation propensity of mutated or modified crystallins associated with cataract formation. These either focus on misfolding-induced aggregation, in analogy to the amyloid formation associated with neurodegenerative diseases[3,7], or suggest that crystallin solubility is lost in the absence of extensive misfolding[8–13]. Depending on the aggregation mechanism, mutated or damaged β/γ-crystallins are predicted to yield protein aggregates with distinct molecular structures. As a result, there is much interest in the structure of aggregated β/γ-crystallins. However, no atomic resolution structural data are currently available for any such aggregate, due to the fact that they are not suitable for solution NMR or X-ray crystallographic study. A further complicating factor is that even the reported morphology of the aggregates varies, a situation similar to the aggregate polymorphism commonly observed among amyloidogenic proteins[22–25]. Two distinct types of crystallin aggregates are widely reported in vitro, which are generally categorized as either 'amorphous' or 'fibrillar'[26–30]. This exact nomenclature, reflecting two distinct types of protein aggregates, is also used for other aggregation-prone proteins[22–25]. Thus, it relates to a more general phenomenon not constrained to cataract-related protein aggregation. Amorphous aggregates are often assumed to be disordered or poorly structured, while the amyloid-like fibrils are typically described as more ordered[26,27]. It is worth noting that the nm-scale morphology of cataract-associated protein aggregates formed in vivo is under debate, although the balance of evidence disfavours a large role for amyloid-like fibrils[2,31,32]. Previously reported arguments in favour of amyloid formation in vivo are largely based on the ability of mouse model cataracts to bind amyloid-binding dyes[31]. However, such dyes are not a foolproof measure of amyloid structure as both false positives and false negatives occur[33,34]. This is borne out by studies that show dye binding by both non-cataract eye lenses and α-crystallins in a non-amyloid state[27,34,35].

To gain insight into the structural features of congenital-cataract-related protein aggregates, we here investigate these aggregates by magic-angle-spinning (MAS) solid-state NMR (ssNMR) spectroscopy and complementary techniques. MAS ssNMR has emerged as the tool of choice for atomic-resolution studies of non-crystalline protein aggregates[36–38]. In particular, we study distinct aggregated states of the above-mentioned P23T hγD that is associated with congenital cataracts (Fig. 1). This mutant protein forms both worm-like fibrils and amorphous-looking aggregates, depending on the experimental conditions. We observe that the former are generated at low pH and display all the characteristics of highly misfolded amyloid-like fibrils. The amorphous-looking aggregates form at physiological pH and lack all of the amyloid hallmarks. Despite their appearance, these aggregates are shown by ssNMR to contain the aggregated protein in a well-ordered native-like conformation, with no evidence of dynamic or static disorder.

## Results

**Polymorphic aggregation of P23T hγD.** First we compare the aggregation that occurs under physiological and denaturing conditions. At physiological pH (pH 7), the 20.6 kDa P23T hγD is soluble at dilute concentrations, but undergoes a temperature-sensitive aggregation at 10–15 mg ml$^{-1}$ concentration (Supplementary Fig. 1). Negative-stain transmission electron microscopy (TEM) reveals the aggregates to be non-fibrillar, with an amorphous appearance (Fig. 2a). Under acidic conditions (pH 3), we observe aggregation of P23T hγD, even at 1.3 mg ml$^{-1}$, which yields 7 nm-wide worm-like unbranched

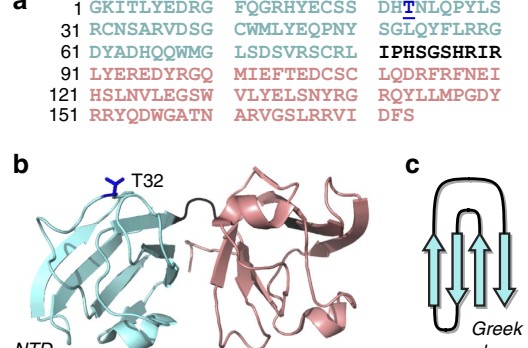

**Figure 1 | Structure of P23T hγD. (a)** Primary sequence and **(b)** solution NMR structure[18]. The mutated residue is underlined and shown in dark blue. The NTD (cyan) and CTD (pink) make up two independently folded symmetric domains, separated by a short linker (black). **(c)** Schematic of the β-sheet-based Greek key motifs present in each domain.

fibrils (Fig. 2b). In fluorescence assays using the amyloid-binding dye thioflavin T (ThT) the two aggregates show a markedly different response (Fig. 2c). Consistent with their fibrillar appearance, the acid-induced aggregates lead to high ThT fluorescence, while the amorphous-looking aggregates do not. In a systematic ThT-binding screen across a broad range of pH values (Fig. 2d), high ThT fluorescence response is observed at both low and high pH, but not under the neutral pH conditions present in eye lenses[39]. ThT assays are not foolproof measures of amyloid structure[33,34]. Therefore, we also performed X-ray powder diffraction measurements on the hydrated aggregates. *Bona fide* amyloids exhibit a characteristic cross-β diffraction pattern, dominated by inter-β-sheet and intra-β-sheet repeat distances of ~4.7 and ~8–10 Å, respectively (for example, see the polyglutamine data in Fig. 2g; ref. 38). This amyloid hallmark is observed for the aggregates formed at pH 3 (Fig. 2e), but is absent for those that form at pH 7 (Fig. 2f). The latter show a different diffraction pattern, featuring 12.6, 16.3 and 28 Å repeat distances. Thus, the fibrils formed on exposure to acidic denaturing conditions have all amyloid hallmarks, but the amorphous-looking aggregates formed under more physiological conditions feature none of them.

**The amorphous-looking aggregates have high internal order.** MAS ssNMR was used to probe the molecular structure of both types of deposits prepared from uniformly ${}^{13}$C and ${}^{15}$N (U-${}^{13}$C,${}^{15}$N)-labelled P23T hγD. First, cross-polarization (CP) ssNMR experiments are used to select for the signals of only the immobilized parts of protein deposits. The one-dimensional (1D) ${}^{13}$C CP spectrum of the pH 7 aggregates is shown in Fig. 3a. The observed peaks have relatively narrow widths, as low as 35 Hz for methyl side chain peaks. More detailed information can be gleaned from two-dimensional (2D) spectra, such as the ${}^{13}$C–${}^{13}$C spectrum in Fig. 3b. In this spectrum, the off-diagonal cross-peaks reflect pairs of carbons that are connected via one or two bonds. Despite the amorphous appearance of the aggregates, the spectrum shows many well-resolved peaks with remarkably narrow peak widths. The narrow linewidths indicate a high degree of structural homogeneity (that is, little structural variation among individual molecules throughout the sample) and a lack of line-broadening dynamics. It appears that the crystal-like structural homogeneity applies to much or all of the aggregated protein, given that these narrow peak widths are seen throughout the spectrum. The 2D spectrum is also characterized by a high degree of peak dispersion, which is normally associated with a well-folded state, especially for proteins with multiple types of secondary structure.

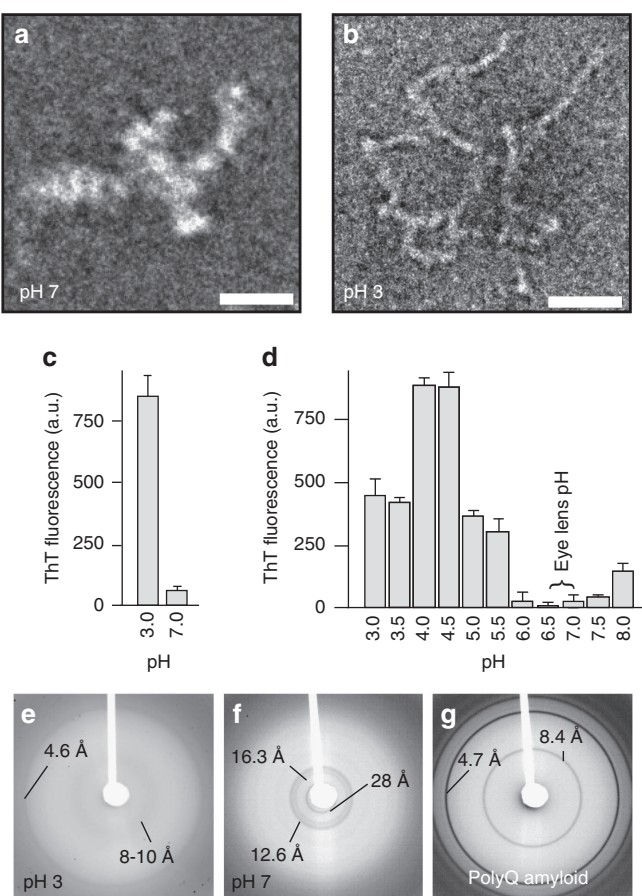

**Figure 2 | Characterization of aggregates formed at 37 °C under different pH conditions.** (**a**,**b**) Negative-stain TEM images of P23T hγD aggregates, formed at pH 7 and pH 3, show amorphous-looking aggregates and worm-like fibrils, respectively. Scale bars, 50 nm. (**c**) ThT fluorescence assay for both of these P23T hγD aggregates. Error bars indicate s.d. (n = 3). (**d**) ThT fluorescence data for 0.13 μM P23T hγD incubated at several pH values. Error bars indicate s.d. (n = 3). (**e**,**f**) X-ray powder diffraction of hydrated P23T hγD aggregates formed at pH 3 and pH 7, respectively. Prominent repeat distances are indicated. (**g**) X-ray powder diffraction of hydrated K₂Q₃₁K₂ polyglutamine fibrils, showing the characteristic amyloid cross-β dimensions of 4.7 and 8.4 Å, as indicated[38].

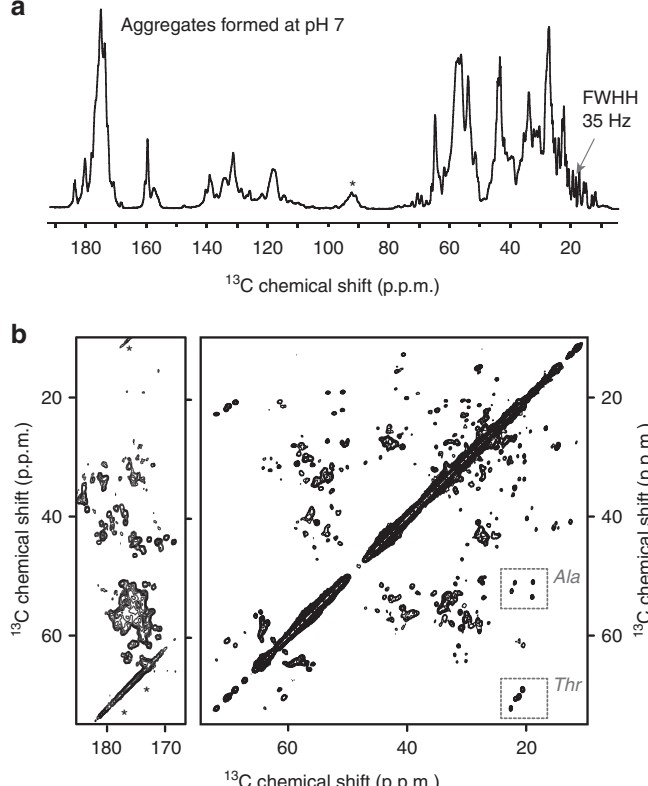

**Figure 3 | MAS ssNMR spectra of uniformly ${}^{13}$C- and ${}^{15}$N-labelled P23T hγD aggregates formed at pH 7.** (**a**) 1D ${}^{13}$C CP spectrum. The width at half height (FWHH) of one of the methyl peaks is indicated. (**b**) 2D ${}^{13}$C–${}^{13}$C ssNMR spectrum with a 8 ms DARR mixing time, resulting in one- and two-bond cross-peaks. Spectra were obtained at 600 MHz (${}^{1}$H) and 12.5 kHz MAS. Spinning side-band peaks are marked with asterisks (*). Dashed boxes mark Ala Cα-Cβ and Thr Cβ-Cγ peaks.

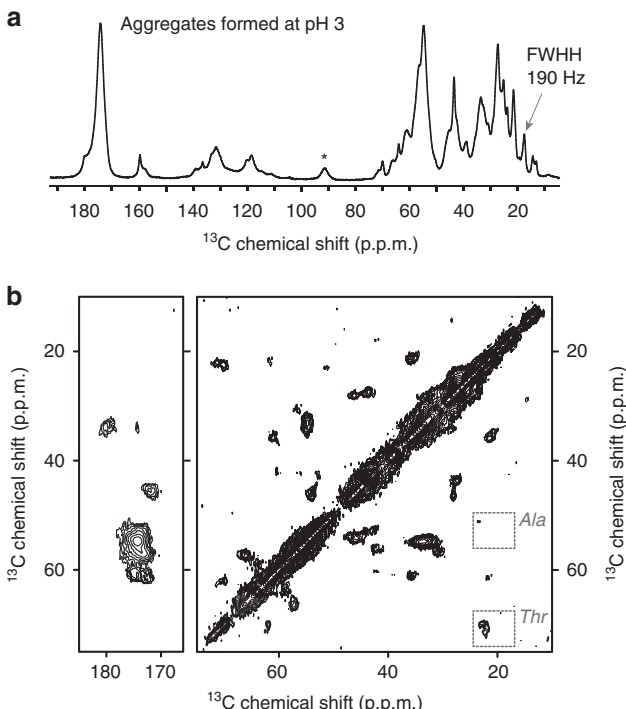

**Figure 4 | MAS ssNMR spectra of uniformly $^{13}$C- and $^{15}$N-labelled P23T hγD aggregates formed at pH 3.** (**a**) 1D $^{13}$C CP spectrum. (**b**) 2D $^{13}$C–$^{13}$C ssNMR spectrum with 8 ms DARR mixing. Both spectra were obtained at 600 MHz ($^1$H) and 12.5 kHz MAS. A spinning side-band peak is marked with an asterisk (*). Ala Cα-Cβ and Thr Cβ-Cγ cross-peaks are enclosed in dashed boxes.

**Solid-state NMR of the acid-induced fibrils.** Analogous ssNMR data were obtained for the acid-induced fibrils. The obtained spectra (Fig. 4) are markedly different, and surprisingly indicate a less ordered internal structure than was present in the amorphous-looking deposits. Figure 4a shows that the 1D spectrum has broader peaks and a limited resolution, which is indicative of a higher degree of structural heterogeneity in these fibrils. The 2D data (Fig. 4b) feature notably fewer and broader peaks compared to the aggregates that form at pH 7. This can be appreciated for instance by comparing the Ala peaks in the boxed regions in Figs 3b and 4b. P23T hγD has four Ala residues that yield four well-resolved peaks in the pH 7 aggregates, but in the fibrils we see a much broader unresolved Ala peak. This observation suggests a lack of chemical shift differences among the observable residues, which points to them being in a similar structure and environment. That said, these CP-based experiments show only the immobilized parts of the protein aggregates; extensive dynamics of parts of the protein could also prevent a subset of residues from being detected.

**Detection of dynamic regions by ssNMR.** To probe for such dynamic protein segments, MAS ssNMR experiments based on the insensitive nuclei enhanced by polarization transfer (INEPT) scheme were also performed. In these INEPT-based ssNMR experiments, only the most dynamic residues are seen, while signals from rigid and immobilized parts of the protein are eliminated[40]. When the same $^1$H–$^{13}$C INEPT experiment is applied to both types of aggregates, we see a striking difference (Fig. 5a). The aggregates formed at pH 3 feature a substantial number of mobile residues, as indicated by the presence of intense peaks in the INEPT spectrum. In striking contrast, the INEPT spectrum of the pH 7 aggregates is devoid of peaks (Fig. 5a; bottom). A 2D $^{13}$C–$^{13}$C

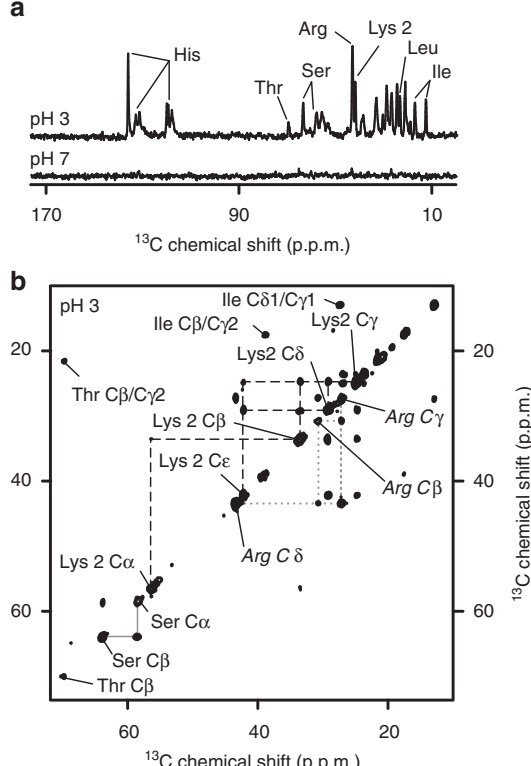

**Figure 5 | MAS NMR spectra of mobile residues within aggregated P23T hγD.** (**a**) 1D $^{13}$C INEPT spectra of U-$^{13}$C,$^{15}$N P23T hγD aggregates formed at pH 3 (top) and at pH 7 (bottom). (**b**) 2D $^{13}$C–$^{13}$C INEPT/TOBSY spectrum of the U-$^{13}$C,$^{15}$N fibrils formed at pH 3. Lys2, Arg and Ser peaks are connected with dashed, dotted and continuous (grey) lines, respectively. Spectra were obtained at 600 MHz ($^1$H) and 8.33 kHz MAS.

INEPT-based ssNMR spectrum for the acid-induced fibrils allows us to probe the identity of the highly mobile amino acids (Fig. 5b). No Ala peaks are detected, indicating a lack of highly mobile Ala residues, which implies that the broad Ala signal in the CP spectrum contains the overlapping peaks from multiple residues in similar conformations. The INEPT spectra feature signals from both backbone and side chain atoms, including those of a Lys residue. Since it is the only Lys in the protein, those mobile Lys signals must stem from Lys-2 near the N terminus of the NTD. Moreover, other visible peaks include Gly, Lys, Ile, Thr and Leu residues, which happen to make up the initial five residues of the NTD (Fig. 1a). Thus, these results are consistent with the initial segment of the NTD being unfolded and flexible, outside the mostly immobilized core of the amyloid fibril assemblies. Additional residues, such as Ser and Arg, are not found at the very N terminus, indicating that other parts of the protein also end up in a similarly dynamic state.

**Secondary structure content of the aggregates.** Solid- and liquid-state NMR resonance frequencies are determined largely by the local bond geometry. The dependence of $^{13}$Cα and $^{13}$Cβ resonance frequencies on the local secondary structure is such that $^{13}$Cα-$^{13}$Cβ peaks occur in distinct spectral regions for residues in random coil, α-helical and β-sheet structures[41]. In Supplementary Fig. 2a–c, we visualize this for some well-resolved amino acids by superimposing the canonical secondary-structure-dependent peak positions on our experimental ssNMR data. Note that these CP-based spectra detect specifically the immobilized parts of the protein aggregates. Independent of the aggregation condition, many of the experimental peaks line up with the

resonance frequencies typical of β-sheets. The dominance of β-sheet structure is most pronounced in the pH 3 fibrils (Supplementary Fig. 2b–d). The spectra of those fibrils lack peaks in the α-helical region, but signals are detected in the intervening spectral region where one expects the signals for residues that lack a well-defined α-helical or β-sheet structure. Interestingly, both Ser (Supplementary Fig. 2a) and Ala peaks (Supplementary Fig. 2c,d) with α-helical chemical shifts are observed in the spectrum of the pH 7 aggregates. Thus, compared to the fibrils, these amorphous-looking aggregates have a more diverse secondary structure content that includes both β-sheet and α-helical elements.

**Spectral modelling of misfolded and native-like aggregates**. As noted above, crystallin aggregation may proceed via mechanisms characterized by different extents of misfolding. Proposed aggregation mechanisms involving extensive un- and misfolding are expected to yield amyloid-like fibrils that feature a β-sheet-based core stabilized by intermolecular hydrogen bonds[7,42,43]. Runaway domain-swapping[8–10] or surface-mediated 'condensation' mechanisms[11–13] would generate aggregates that retain much of the native fold and lack an amyloid core[37,44]. To evaluate and visualize the expected spectral differences between misfolded amyloid and native-like aggregates, we generated schematic simulated $^{13}$C–$^{13}$C ssNMR spectra. To coarsely model a typical amyloid structure, we applied canonical parallel β-sheet torsion angles ($\phi,\varphi = -119°, 113°$) to the P23T hγD primary sequence. This backbone structure model was then used to predict the $^{13}$C′, $^{13}$Cα and $^{13}$Cβ chemical shifts, which were used to generate the $^{13}$C–$^{13}$C peak patterns in Fig. 6a. To simulate the ssNMR spectrum of native-like aggregates, the NMR shifts of the soluble protein were used as a starting point[18]. These solution NMR shifts by definition correspond to the soluble protein's native fold, and led to the simulated spectrum in Fig. 6b. The two simulated spectra illustrate the marked qualitative differences expected for amyloid-like fibrils and native-like aggregates. The former are characterized by a lack of peak dispersion for residues in the purely β-sheet amyloid core. With their mixed secondary structure, the native-like aggregates have a much larger peak dispersion that resolves many of the individual amino acids.

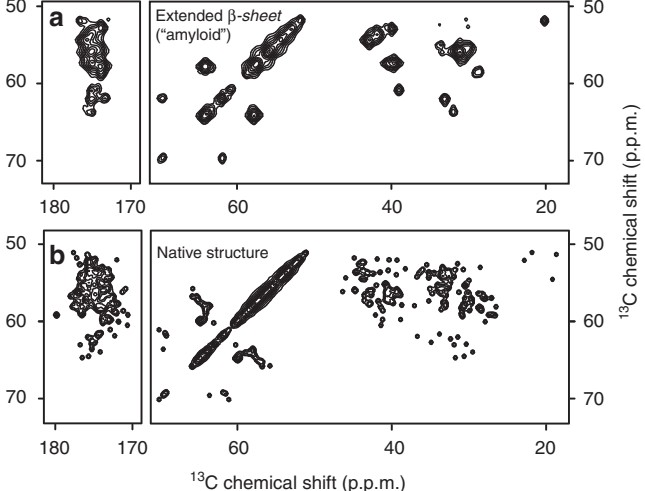

**Figure 6 | Synthetic $^{13}$C–$^{13}$C ssNMR spectra predicted for aggregates containing either typical amyloid-like misfolded proteins or natively folded P23T hγD.** (**a**) Simulated spectrum predicted for P23T hγD in a parallel in-register β-sheet structure, as is typical for amyloid-like fibrils. (**b**) Simulated spectrum generated from the solution NMR chemical shifts of natively folded P23T hγD[18].

**Acid-induced P23T hγD fibrils lack native-like structure**. The experimental ssNMR data for the acid-induced fibrils (Fig. 4) match well to the amyloid-like fibril model, while showing little to no correspondence to the spectra expected for native-like aggregates. As noted above, the experimental spectrum features some peaks from non-β residues that do not appear in the synthetic spectrum. We attribute these peaks to loops and other segments that are outside the β-sheet amyloid core, but are immobilized enough to be seen in the CP-based spectra, similar to the non-amyloid flanking domains of huntingtin exon 1 fibrils[45]. Such non-core residues were not represented in our simplistic amyloid model designed to predict the spectral features of a canonical amyloid core architecture.

**The amorphous aggregates have a native-like structure**. In contrast to the acid-induced fibrils, the spectrum of amorphous-looking aggregates obtained at pH 7 (Fig. 3) shows a striking resemblance to the simulated spectrum expected for native-like aggregates. To provide more detailed insights into the structural similarities and differences between the aggregated and soluble state, Supplementary Fig. 3 overlays the 2D $^{13}$C–$^{13}$C ssNMR spectrum of P23T hγD aggregated at pH 7 with peak markers representing the solution NMR assignments. Even in absence of site-specific assignments (see below), a close peak-by-peak inspection of this single 2D spectrum reveals that the majority of peak positions are remarkably similar in the solid and solution states. A limited number of well-resolved solution peak markers do not overlap with a matching ssNMR peak, as indicated with circled crosses in Supplementary Fig. 3. Conversely, a subset of ssNMR peaks do not have a matching solution NMR resonance (marked with coloured lines in Supplementary Fig. 3). Assignment experiments (below) show that the 'missing' solution NMR peaks and 'unmatched' ssNMR peaks represent the same specific residues. Thus, we are not seeing a disappearance of peaks, but rather a changing of NMR resonance frequencies. This is consistent with the fact that no highly flexible residues (which would be invisible in the CP spectra) were detected in the pH-7 aggregates (Fig. 5). Residue-specific assignments of several key segments of the aggregated protein were obtained based on a backbone-walk analysis of 2D and three-dimensional (3D) homo- and heteronuclear ssNMR experiments (Supplementary Fig. 4; Supplementary Table 2). Comparing the available solid- and solution-state chemical shifts, Supplementary Fig. 4c shows the residue-specific chemical shift perturbation (CSP) that accompanies the formation of the amorphous aggregates. These data support the visual analysis of the 2D $^{13}$C–$^{13}$C data above, showing that many residues have a CSP similar to the uncertainty in the chemical shifts (the average $^{13}$C CSP is 0.55 p.p.m.). The largest $^{13}$C chemical shift deviations are ~2 p.p.m., but this represents only a few residues among those that were assigned: G1, T23, I81 and I170. The chemical shift differences between the pH 7 aggregates and the hypothetical amyloid-like β-sheet model from Fig. 6a are more than twice as large (Supplementary Fig. 5). This overall pattern is illustrated in Fig. 7a,b, based on the well-resolved Ala and Thr signals. Unlike the acid-induced fibrils (bottom panels), the amorphous-looking aggregates yield Ala peaks that closely resemble the solution NMR shifts (marked with X). Given the Ala distribution throughout the protein (Fig. 7c), this similarity in itself strongly suggests a retention of much of the native fold. Among the four Thr residues, two (T23 and T159) have significantly changed chemical shifts, indicating a localized change in structure. Returning to a more global view of the protein aggregates formed at pH 7, Fig. 7d highlights those residues for which we observe significant (> 0.5 p.p.m.) $^{13}$C shift changes between the solution and solid-state NMR data. The

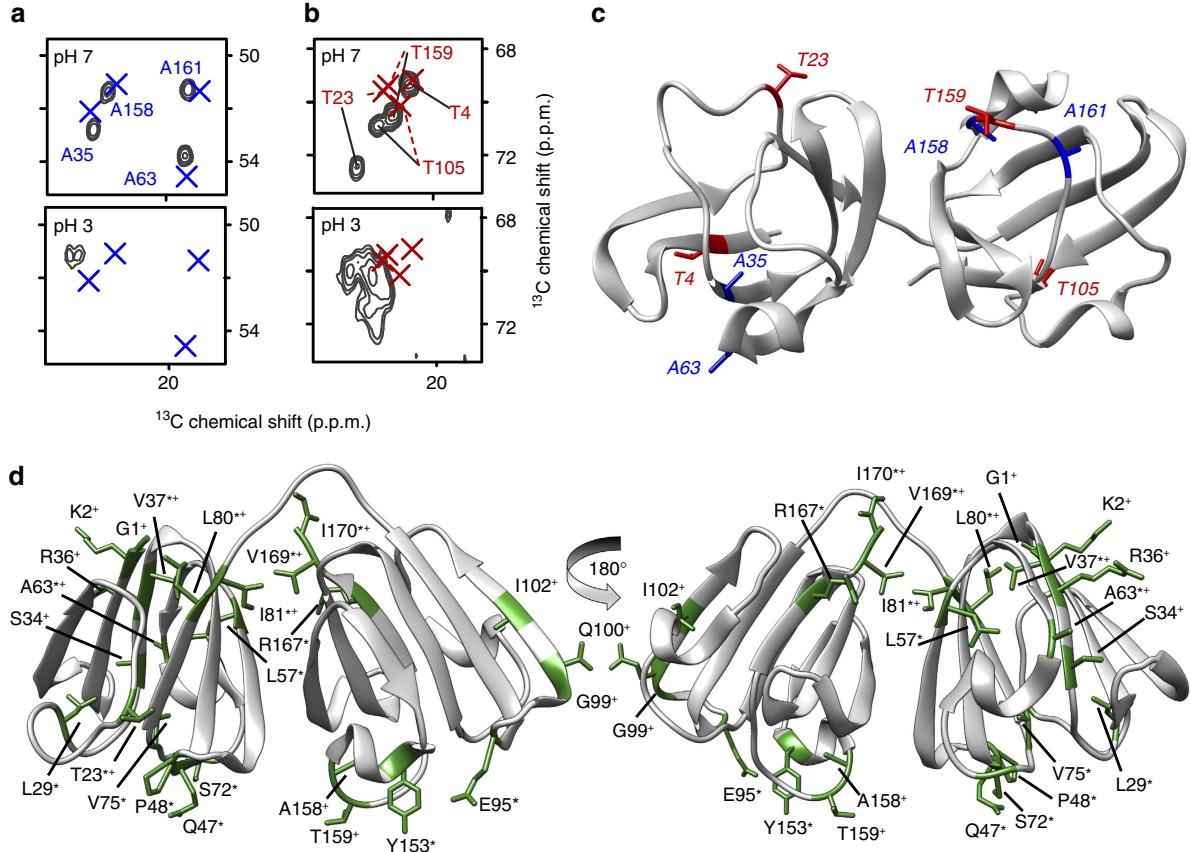

**Figure 7 | Mapping of NMR-detected structural changes and similarities.** (**a**) Ala Cα-Cβ cross-peaks for the aggregates obtained at pH 7 (top) and pH 3 (bottom). Experimental ssNMR cross-peaks are depicted in dark grey and blue crosses mark the solution NMR shifts of the soluble protein[18]. (**b**) Analogous depiction of the Thr Cβ-Cγ cross-peaks. (**c**) P23T hγD structure[17] highlighting the locations of all eight Ala and Thr residues. (**d**) Structural mapping of residues whose NMR resonances change on aggregation at pH 7 (Supplementary Figs 3 and 4) due to aggregation-associated changes in the local structure or environment. Residues whose resonances are different between the solution and aggregated (pH 7) states are labelled in green on the ribbon diagram. Residues marked with '+' have unambiguous ssNMR assignments (Supplementary Table 2); those marked with '*' were identified in the visual analysis of Supplementary Fig. 3.

indicated residues are identified either based on *de novo* ssNMR assignments (Supplementary Table 2; marked with + in Fig. 7d), the spectral analysis of Supplementary Fig. 3 (*), or both. This representation indicates that residues that sense a structural change upon aggregation are mostly outside the core β-strand segments of the Greek key motifs and are found instead on the surface-exposed regions of the protein's folded subdomains.

## Discussion

We observed the aggregation of P23T hγD across a range of pH conditions, and found that the nature of the resulting aggregates is different at different pH values. The aggregates formed at non-physiological pH values showed high fluorescence in assays using the amyloid-binding ThT dye. The amyloid-like nature of the acid-induced aggregates was validated by their fibrillar morphology and characteristic cross-β X-ray diffraction pattern (Fig. 2e). As is typical for amyloid-like fibrils, the ssNMR peak patterns of the fibrils differed markedly from those expected for the protein's native fold. The protein had undergone extensive restructuring during aggregation at pH 3. The ssNMR data we have obtained thus far are unable to distinguish the different kinds of β-sheet amyloid core architectures[43]. That said, as previously suggested[42], it seems likely that the fibril core would feature the prototypical parallel in-register β-sheet structure most commonly seen in other amyloid-like fibrils[7]. The ssNMR spectra contained signals

from residues lacking β-sheet structure that are either dynamically (Fig. 5b) or statically disordered (Supplementary Fig. 2). The amino acids are presumably located outside the β-sheet-based amyloid core. Thus, our data support a model featuring fibrils built around a β-sheet amyloid core that is decorated with disordered non-β segments. This structural motif qualitatively resembles the fibril assembly model described in prior infrared studies of hγD aggregated under acidic conditions *in vitro*[42].

Consistent with prior studies[19,20,46], we found that denaturing conditions were not necessary for P23T hγD to aggregate at concentrations well below the high solubility of WT hγD. The P23T hγD aggregates formed at physiological pH values lack the characteristic cross-β signature of amyloids, have low ThT fluorescence, and have an amorphous appearance by TEM. A similar amorphous morphology was previously noted for P23T hγD aggregated under similar conditions[20], and also for other β- and γ-crystallins[28-30,47]. As noted in the opening paragraph, the same nomenclature is used to describe non-fibrillar aggregates formed by other aggregation-prone proteins[22-25]. Prior studies offer little insight into the internal structure of these deposits, which are often described as 'unstructured'[20,26,27] or 'disordered'[21,27,29,48]. Our ssNMR studies paint quite a different picture. Both the dispersion and widths of our ssNMR peaks are reminiscent of those typically seen for crystalline protein preparations[49]. This means that the individual protein

molecules in the sample all share a single, well-defined conformation. In CP-based spectra of the immobilized parts of the aggregates we observed residue counts that match the solution NMR data, with no sign of missing peaks. This was reaffirmed by an INEPT spectrum that was devoid of 'mobile' signals (in contrast to the acid-induced fibrils). Thus, the seemingly amorphous deposits are actually highly homogeneous assemblies of well-structured protein monomers on the microscopic level, with no evidence of either dynamic or static disorder. Supplementary Fig. 3 shows the striking similarity between the 2D ssNMR spectra of the pH 7 aggregates and the known chemical shifts of the natively folded state present in solution. Residues featuring atoms with a significant change in their chemical shifts were found to be outside the β-strand segments of the native fold's Greek key motifs (Fig. 7d; Supplementary Fig. 4). These sites of structural change likely co-localize at least in part with the intermolecular contacts that arise on aggregation, but are absent in the monomeric soluble protein. The limited spectral changes detected in the aggregates' ssNMR data are incompatible with large-scale un- or misfolding.

The fact that the ssNMR spectra of the pH 7 aggregates indicate a native-like structure is most consistent with two types of aggregation mechanisms proposed in the literature. 'Condensation' mechanisms suggest that minor changes in surface charges or surface hydrophobicity would be sufficient to increase the molecule's aggregation propensity, without substantial mis- or re-folding (Fig. 8b)[11–13,50]. Even relatively small changes in surface characteristics may disrupt the careful balance of inter-protein interactions that normally facilitate the remarkable protein concentrations in the eye lens. The proteins could then aggregate while preserving a largely native fold, with the mutations affecting the surface patches that mediate the protein–protein contacts in the aggregated state. Figure 7d summarized the identities and locations of residues for which we observe above-average chemical shift deviations between the soluble and aggregated protein (Supplementary Figs 3 and 4). These are the residues that experience a change in local structure

as a result of the aggregation process. They are found to reside outside the core β-strands of the NTD and CTD, and are instead primarily on the surfaces of the domains, indicating an apparent preservation of the Greek key motifs.

Deposits assembled from intertwined or 3D domain-swapped proteins, as previously proposed for cataract-associated aggregates[8–10], would also be predicted to have native-like spectra[37]. On the basis of known domain-swapped multimers as large as tetramers, it has been proposed that 'runaway' domain swapping in a daisy-chain manner could be a mechanism of protein aggregation. A hypothetical domain-swapped P23T hγD assembly is shown in Fig. 8d. We stress that this is a hypothetical graphic, purely meant to offer a schematic visual representation of the structural implications of runaway domain swapping. The swapped multimers reproduce much of the native fold, with structural changes centred on loop regions that act as hinges between the swapped domains. A characteristic feature of domain-swapped assemblies would be that notable NMR chemical shift changes would primarily cluster in such a hinge-loop region. The observed ssNMR shift differences (Fig. 7d) fail to match this particular pattern, leaving us unable to identify a specific hinge-loop region, or build a particularly convincing runaway domain-swapped model. As a result, we interpret the available data to be most consistent with the kind of aggregates expected for a condensation-type aggregation process. That said, there are many different ways in which a domain-swapped assembly could be created, such that it is difficult to exclude a role for domain swapping until more structural data are available.

Various γ-crystallins form amyloid-like fibrils in vitro, at acidic pH (refs 13,30,42,51) or when exposed to other denaturing conditions[8,13,28,31,32,52–54]. A similar behaviour has been reported for α-and β-crystallins[13,34,54]. However, numerous globular proteins can be induced to form amyloid-like fibrils when one destabilizes their native fold, independent of their propensity to form amyloid under physiological conditions[3,37]. In the case of P23T hγD, the mutation itself does not cause much change in the structure, as mentioned in the opening paragraphs and discussed

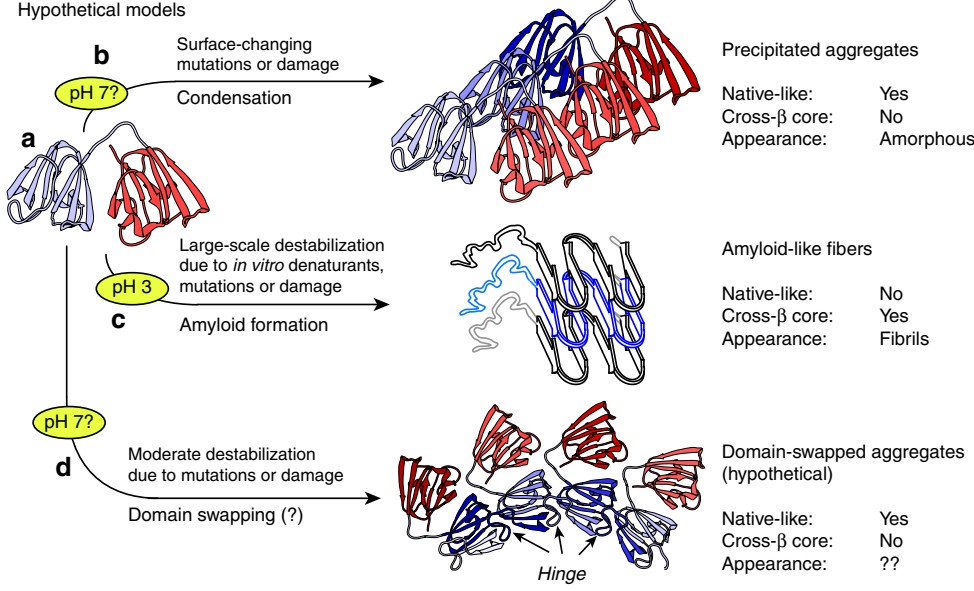

**Figure 8 | Proposed potential aggregation mechanisms.** (**a**) The native hγD fold, with the NTD and CTD in blue (left) and red (right), respectively. (**b**) Condensation into aggregates within which individual proteins retain a native-like fold. Mutations or chemical damage may lead to changes in surface charges or hydrophobicity, thus disrupting the native protein–protein interactions essential for high solubility. Individual protein monomers are depicted in different colour shades. (**c**) Denaturing conditions, including low pH, induce a loss of the native fold and formation of fibrils with a β-sheet-based amyloid core. Unfolding may also result from disruptive mutations or chemical damage to buried residues. (**d**) Hypothesized domain-swapping-based mechanism that generates aggregates in which much of the native fold is preserved, except for loops that act as hinges and undergo substantial rearrangements.

in detail in prior studies[17,18]. Additional destabilization of the native fold, for instance by exposure to acidic or basic pH values, appears to be necessary for amyloid formation to become the dominant aggregation pathway (Fig. 8c). Thus, it is tempting to conclude that P23T hγD aggregation under *in vivo* conditions is likely to occur via a non-amyloid mechanism. Of course, extrapolation to the complex *in vivo* conditions is always difficult, given the potential impacts of other β/γ-crystallins, α-crystallin chaperones and the build-up of ageing-induced chemical modifications[13,29]. That said, there is one respect in which cataract-related protein aggregation may be fundamentally different from amyloid formation in neurodegenerative disorders. In the latter case, the amyloid-forming proteins or peptides are often present at very low concentrations[55,56]. The concentration of protein in the eye lens is orders of magnitude higher, requiring the remarkable eye lens protein solubility that is based on a careful balancing of repulsive and attractive protein–protein interactions[57]. Amyloidogenic aggregation at low concentrations tends to involve exposure of hydrophobic sites through misfolding, and proceed via specific nucleation events[38,58]. At very high-protein concentrations, aggregation can result from surface-mediated changes in protein–protein interactions, well before large-scale structural changes trigger the formation of an amyloidogenic state.

Protein aggregation processes involved in cataract formation may be qualitatively different from the misfolding-based amyloid formation associated with neurodegenerative disease. Recent efforts to develop and screen anti-cataract drugs have focused on their inhibition of amyloid formation[53,54]. Our results argue that it may be important to also screen against disease-related crystallins that form non-amyloid aggregates. We showed that the amorphous-looking aggregates are much more ordered and structured than one might have expected based on their morphology. As such, at least in this case, the 'amorphous' label appears to be a misnomer when it comes to the atomic level structure of the aggregated protein. One implication of this remarkable structural order and homogeneity is that the deposits presumably form via a reproducible and specific aggregation mechanism, rather than a multiplicity of parallel or non-specific pathways. A better molecular understanding of this aggregation process may pave the way for the development and design of small-molecule aggregation inhibitors[59,60]. Age-dependent changes in the operation of nature's own aggregation inhibitors, protein chaperones, play a key role in ageing-dependent aggregation diseases. In the eye, the α-crystallins perform this protective role. Intriguingly, α-crystallins act on amorphous-looking and amyloid-like substrates through different mechanisms[4,5]. Further structural studies of the crystallins' polymorphic aggregates will enhance our understanding of these protective processes and may enhance ongoing efforts to develop new preventative or curative treatments.

## Methods

**Expression and purification of P23T hγD.** The mutant P23T hγD gene (*CRYGD*) was cloned into a pET14b vector[18]. Transformed One Shot *Escherichia coli* BL21star (DE3) cells were grown in LB broth or modified M9 minimal medium at 37 °C to an OD$_{600}$ of 0.6, induced with 1 mM IPTG for protein expression at 37 °C for 4 h. Isotopically labelled protein was obtained by growing cells in modified M9 minimal medium containing 2 g l$^{-1}$ U-$^{13}$C-D-glucose and 1 g l$^{-1}$ $^{15}$N-ammonium chloride (Cambridge Isotope Laboratories, Tewksbury, MA). Cells were harvested by centrifugation and cell pellets were resuspended in QA buffer (50 mM Tris at pH 8.2, 1 mM EDTA, 1 mM dithiothreitol (DTT)), and lysed via microfluidization (Microfluidics, Newton, MA). Nucleic acids were removed from the cell lysate by precipitation with 1% (w/v) polyethyleneimine, and cell debris was removed by centrifugation at 39,000 *g* for 20 min (Sorvall RC 3C Plus, H-6000A rotor). Proteins were purified using a multi-step protocol[16,18]. Clarified cell lysate was loaded onto a pre-equilibrated HiTrap Q XL anion exchange column (GE Healthcare) in QA buffer, and the flow-through fraction was collected and

dialysed overnight at 4 °C into SA buffer (10 mM MES at pH 6.0, 1 mM EDTA, 1 mM DTT, 2% v/v glycerol). After the dialysis any precipitates were removed by centrifugation at 39,000 *g* for 20 min. The supernatant was loaded onto a pre-equilibrated HiTrap SP cation exchange column (GE Healthcare) in SA buffer and eluted using a linear gradient of 0–1 M NaCl. Fractions containing P23T hγD were collected and further purified by gel filtration on a Superdex 75 26/60 column (GE Healthcare) in 100 mM sodium phosphate buffer (pH 7), containing 5 mM DTT and 0.02% (w/v) NaN$_3$. The concentration of the final purified protein was determined with a NanoDrop 1000 spectrophotometer (Thermo Scientific) using an A$_{280}$ molar extinction coefficient of 42.86 mM$^{-1}$ cm$^{-1}$.

**Preparation of P23T hγD aggregates.** To prepare aggregates under acidic conditions, 10% and then 1% (w/w) HCl was added drop-wise to 1.28 mg ml$^{-1}$ (62 μM) solutions of P23T hγD on ice, until the pH reached 3.0 (ref. 42). The sample was immediately vortexed after the addition of each HCl droplet to insure proper mixing and also to avoid any local areas of low pH within the sample. After adjusting the pH to 3.0, the protein was allowed to aggregate overnight or longer, at 37 °C. To prepare aggregates under neutral conditions (pH 7), P23T hγD was concentrated to >10 mg ml$^{-1}$ (0.49 mM) over four 10-min and one 5-min steps using 3 kDa Centriprep centrifugal filters spinning at 3,200 *g* (Eppendorf 5810R, A-4-81) at 4 °C in 100 mM sodium phosphate buffer (pH 7) containing 5 mM DTT and 0.02% (w/v) NaN$_3$. The protein concentration was measured occasionally using ultraviolet absorbance until the sample appeared cloudy by eye (Supplementary Fig. 1), or reached the desired concentration. Subsequently, the protein was allowed to aggregate overnight at 37 °C (ref. 20). [U-$^{13}$C-$^{15}$N]-labelled P23T hγD aggregates were formed using the same conditions and concentrations (in mg ml$^{-1}$) used to form the unlabelled aggregates.

**Transmission electron microscopy.** Aggregates of P23T hγD formed under acidic (pH 3) and neutral (pH 7) conditions were imaged using TEM. Samples (5 μl) were adsorbed on freshly glow discharged 400 mesh size carbon-coated copper grids for 15 s, and the excess was removed by blotting with filter paper. The sample grids were subsequently stained with 1% (w/v) sodium phosphotungstate for 5 s, and blotted. Grids were imaged at 11,000-fold (pH 7) and 21,000-fold (pH 3) magnification using a Tecnai T12 transmission electron microscope (FEI; Hillsboro, OR) operating at 120 kV and equipped with an UltraScan 1000 CCD camera (Gatan; Pleasanton, CA).

**X-ray powder diffraction.** Aggregates of P23T hγD formed under acidic (pH 3) and neutral (pH 7) conditions were pelleted by centrifugation. Aggregates formed under neutral conditions were pelleted at 14,000 *g* for 20 min in a table-top VWR Galaxy 14D centrifuge, while aggregates formed under acidic conditions were pelleted by ultracentrifugation at 100,000 *g* for 4 h (Beckman Coulter Optima MAX Ultracentrifuge, with TLA 120.2 rotor). The majority of the supernatant was removed, and the pellet was resuspended in the remaining buffer and packed into a glass capillary (0.7 mm) using a syringe. Capillaries were sealed with wax to retain hydration of the samples. Diffraction data were measured at room temperature with a Rigaku Saturn 944 CCD camera (Tokyo, Japan).
A Rigaku FR-E generator (2 kW, spot size 0.07 mm) was used as the X-ray source. Aggregates formed under neutral conditions were exposed for 75 s, while aggregates formed under acidic conditions were exposed for 210 s. Diffraction data were processed and analysed in Structure Studio (Rigaku).

**Thioflavin T fluorescence measurements.** Suspensions of ∼25 μg of aggregates formed under acidic (pH 3) and neutral (pH 7) conditions were diluted 20× to 200× (depending on the pre-aggregation concentration) into a ThT stock solution (50 μM ThT, 10 mM sodium phosphate, 150 mM NaCl, pH 7). Samples were excited at 445 nm and fluorescence emission was recorded at 489 nm using a FluoroMax-4 Research Spectrofluorometer (Horiba; Kyoto, Japan). Excitation and emission slits were 2 and 4 nm, respectively. In addition, ThT fluorescence measurements were also performed for a series of P23T hγD samples incubated at pH 3 to pH 8 in 0.5 pH increments. To prepare these aggregates, the pH of a low-concentration solution of P23T hγD (0.13 mg ml$^{-1}$) was adjusted using 10 and 1% (w/w) HCl and 0.5 M NaOH. Samples were allowed to aggregate at 37 °C for 5 days, after which ∼2.5 μg (20 μl) of the resuspended P23T hγD aggregates were diluted to 400 μl in the same ThT buffer mix, for each ThT fluorescence measurement.

**MAS ssNMR spectroscopy.** Hydrated [U-$^{13}$C-$^{15}$N]-labelled P23T hγD aggregates formed under acidic (pH 3) and neutral (pH 7) conditions were separately packed into regular- and thin-walled 3.2 mm zirconia MAS rotors (Bruker Biospin, Billerica, MA). The sample packing was done by pelleting the hydrated sample suspension directly into the MAS rotor using an ultracentrifugal packing device[61] under centrifugation at ∼130,000 *g* for 2–3 h in a Beckman Coulter Optima L-100 XP ultracentrifuge with a SW-32 Ti rotor. Excess supernatant was removed, the fully hydrated sample was sealed, and subsequently studied by ssNMR in an unfrozen and hydrated state. All MAS ssNMR experiments were performed on a widebore Bruker Avance I NMR spectrometer operating at a $^1$H Larmor frequency

of 600 MHz (14.1 T) using a 3.2 mm MAS NMR probe with an HCN 'EFree' coil (Bruker Biospin). A constant flow of cooled gas was used to control the sample temperature. Spectra were acquired with Bruker Topspin software, processed in NMRPipe, and analysed with Sparky software and the CcpNmr Analysis program developed by the Collaborative Computation Project for the NMR community (CCPN)[62,63]. A 90° shifted sine bell followed by a Lorentzian-to-Gaussian transformation function was applied in all three dimensions of the NCACX and NCOCX spectra during processing. $^{13}C$ and $^{15}N$ chemical shifts were indirectly referenced to 4,4-dimethyl-4-silapentane-1-1 sulfonic acid and liquid ammonia, respectively, based on external reference measurements of the $^{13}C$ signals of adamantane[64]. CP spectra were used to study the immobilized parts of the protein aggregates. Any unaggregated proteins would not be visible in such CP experiments, such that the signal must be from the aggregated protein. 1D $^{13}C$ CP spectra were acquired at a 12.5 kHz MAS rate using ramped $^1H$–$^{13}C$ CP. Intra-residue through-space $^{13}C$–$^{13}C$ correlations were obtained from 2D experiments that combined ramped $^1H$–$^{13}C$ CP with 8 ms dipolar-assisted rotational resonance (DARR) $^{13}C$–$^{13}C$ mixing[65]. Partial assignments (Supplementary Table 2) were obtained for the aggregates formed under neutral conditions, with the aid of the $^{13}C$–$^{13}C$ spectra as well as 2D and 3D $^{15}N$–$^{13}C$ correlation experiments (NCO, NCA, NCOCX, NCACX), measured at 10 and 12.5 kHz MAS. These experiments involved 1.5 ms of $^1H$–$^{13}N$ CP, 5.0 ms of ramped $^{13}C$–$^{15}N$ SPECIFIC CP (spectrally induced filtering in combination with CP)[66] and 15.0 ms of $^{13}C$–$^{13}C$ DARR mixing, where applicable. The rotor-synchronized refocused INEPT scheme was used to acquire 1D J-coupling-based $^{13}C$ spectra at a 8.333 kHz MAS rate[67]. These experiments employed $\tau_1 = 1.2$ ms and $\tau_2 = 1.0$ ms for aggregates formed at neutral pH, and $\tau_1 = 1.5$ ms and $\tau_2 = 1.0$ ms for the fibrils formed at pH 3. 2D spectra showing through-bond $^{13}C$–$^{13}C$ correlations between highly mobile carbons were obtained using a combination of $^1H$–$^{13}C$ INEPT and 6.0 ms total through bond correlation spectroscopy (TOBSY) $^{13}C$–$^{13}C$ mixing[68]. Two-pulse phase modulation $^1H$ decoupling of typically 83 kHz was applied during acquisition for all 1D experiments, and during acquisition and evolution for 2D experiments[69]. Additional experimental details can be found in Supplementary Table 1 in the Supporting Information.

**Synthetic MAS ssNMR spectra.** Synthetic NMR spectra were generated for both native and amyloid-like reference structures using methods similar to those used in our prior work[70]. Spectra representative of the native protein structure were generated starting with the published solution NMR chemical shifts of soluble P23T hγD (ref. 18). Alternatively, a hypothetical completely parallel β-sheet structure of P23T hγD was built using the PyMOL Molecular Graphics System (Schrödinger, LLC), from which chemical shifts were predicted using the SPARTA+ program[71]. Subsequently, synthetic 2D NMR spectra showing the predicted one-bond cross-peaks were generated using utilities from the NMRPipe software package[62]. The solution NMR peak markers in Supplementary Fig. 3 were generated from the solution NMR chemical shifts[18] using the synthetic peak list creation routines of the CcpNmr Analysis program[63].

**Data availability.** The ssNMR chemical shifts of P23T hγD aggregated at pH 7 that were assigned are available in Supplementary Table 2, and as ID 27039 in the BMRB database. The data that support the findings of this study are available from the corresponding author on request.

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

## Acknowledgements

We thank Drs Cody Hoop, Jinwon Jung and Fangling Ji for helpful discussions and for performing early experiments related to the current work, and Abhishek Mandal for help with experiments and helpful discussions. We acknowledge Michael Delk, Doowon Lee and Alexander Makhov for assistance with NMR, X-ray, and EM instrumentation. We thank Dr James Conway for use of the departmental EM facility. This work was enabled by funding from the University of Pittsburgh, the National Institutes of Health (R01 EY021193 to A.M.G, R01 GM112678 to P.C.A.v.d.W., T32 GM088119 to J.C.B.), the Achievement Rewards for College Scientists (ARCS) Foundation (J.C.B.), and grant UL1 RR024153 from the National Center for Research Resources (NCRR). Molecular graphics were prepared with UCSF Chimera, developed by the Resource for Biocomputing, Visualization, and Informatics at the University of California, San Francisco (supported by NIGMS P41-GM103311).

## Author contributions

J.C.B. and P.C.A.v.d.W. conducted the ssNMR experiments. J.C.B., M.L. and P.C.A.v.d.W. analysed ssNMR data. J.C.B. and M.J.W. prepared samples. J.C.B. performed the TEM, X-ray, ThT and aggregation experiments. J.C.B., P.C.A.v.d.W., M.J.W. and A.M.G. designed the experiments; J.C.B. and P.C.A.v.d.W. wrote the paper. All authors reviewed and edited the manuscript and approved its final form.

## Additional information

**Competing interests:** The authors declare no competing financial interests.

