## [Peer Review File · Nature Communications]

Reviewers' Comments:

Reviewer #1 (Remarks to the Author)

This is a great paper. The authors show quite convincingly, by ThT fluorescence, electron microscopy, and especially solid state NMR, that gamma-D-crystalline aggregates that form at physiological pH are definitely not amyloid-like. The data are clearly presented and solid, and interpreted appropriately. I almost never recommend in favor of publication without revisions, but in this case I must make an exception. This paper should be published in its current form.

Reviewer #3 (Remarks to the Author)

The groups of van der Wel and Gronenborn present an interesting study of gammaD crystallin (GDC for this review) that potentially overturns a paradigm in the field of amyloids. The results are highly significant and the main point of the paper is hard to miss. Namely, at pH 7 the P23T mutant of GDC forms aggregates that look amorphous by microscopy but in fact have very high microscopic order as evidenced by the ssNMR spectra, but at lower pH the aggregates look more like typical amyloid based on the low resolution measures, yet have very poor microscopic order based on the ssNMR. The majority of the paper then is devoted to modeling these ideas and postulating on the mechanisms.

Unfortunately, there are quite a few data sets and discussion points left out that one would need to make this study complete.

1. From the beginning, the structure of P23T GCD is discussed but not the wild type. Presumably there is a difference (albeit minor) in the structure that impacts the behavior of the mutant in forming aggregates. At figure 1 it would be helpful to show the differences in the monomer structure, and later to include the WT chemical shifts in the comparison modeling.
2. I am not sure the term “amorphous” should be used in the title and/or elsewhere since in fact this is a misnomer. Can the authors please address this point more explicitly somewhere?
3. A somewhat major concern is that the spectra are not assigned, although they could be. The 2D CC spectrum is very well resolved and many residue types give outlier CA/CB shifts; the CA-CO region is nicely separated and most of the Ala and Thr and Ser residues give clean peaks in the 2D. To perform now-standard NCACX and NCOCX spectra even as 2Ds would likely confirm the specific assignments. Much of the paper relies on comparison of modeled spectra with experimental spectra, and hypothesized assignments are indicated on several of the figures but it is not clear whether any of these signals were actually confirmed. If not, then (for example) in Fig 7 and S2 the labels should be removed or at least a caveat placed in the caption. Certainly

if residues T23, A63, I170 etc. have been identified with confidence, the logic should be elaborated upon somewhere. But the better solution would be to have the backbone correlations.

4. Related to the point above, clearly the modeling gives a nice result but it is hard to assess the relative quality of the various models without quantitation. Methods are available for quantifying the agreement between models and experimental data of this type. It would strengthen the argument to provide such quantitation. I am sure the NMR experts see the agreement but maybe not so much the non-experts, especially since quite a few of the expected peaks are missing and/or slightly shifted. How would this compare with the WT shifts for example?

5. Around lines 205-208 specifically the discussion is weak due to a lack of specific assignments. Resonance frequencies are changing, but for which residues? Again it is my view that these assignments can and should be made to strengthen the arguments here.

6. Again around line 227-230 there is discussion that the ssNMR data “are unable to distinguish the different kinds of beta-sheet amyloid core architectures”. At minimum the authors should add qualifiers to this statement (“The ssNMR data we have obtained so far do not yet distinguish...”). But even so, the assignments alone would distinguish the cases proposed on the basis of chemical shift perturbations relative to the solution data perhaps. Also a single long mixing time CC 2D spectrum would give cross peaks, very likely which would confirm or deny the postulated models. Mixed labeling would also inform upon this. So the authors could discuss these options and preferably in my view perform these experiments to test the hypothesis.

7. Again it is not clear (line 270) what is meant by chemical shift changes. Which surface-facing residues? Are these site-specifically assigned?

8. Around line 276 there is stress on the hypothetical nature of the structure models. This should also be made plain on the figure caption and figure.

9. Line 292 mentions for the first time in this paper that the P23T “mutation itself does not cause much change in the structure”. First, what does “much change” mean here? Second, as noted already above

10. I missed the experimental data for the WT protein. Was there any? Is the behavior at all unique to the P23T mutant? Would the same native fold aggregate be obtained from WT under these conditions? This seems to be a very important question to answer since the title implies unique behavior of this mutant. Exploring other mutants would be less essential, but a comparison with WT seems to be a fundamental part of the paper that is missing. (at least kinetic data such as in Fig S1)

11. line 360: so if the concentration is increased up to the point that the solution becomes cloudy, without a defined end point of concentration based on the physiological case, then it would be expected that the WT would do the same thing but perhaps at a slightly higher concentration (according to Dobson all proteins do this eventually)

Minor points:

1. p. 7 line 111: "So-called" cross polarization?
2. p. 8 lines 148-149: Most Lys sidechains are mobile even if the backbone is not, so it is unclear that this is proof the NTD being highly mobile.
3. Fig 7d has an incorrect label (L23, but it must be T23). Please check the others too.
4. line 337: Tewksbury? (or Andover?)
5. line 357: why not mix continuously?
6. Table S1 gives all the information needed to compute the total experimental time but it would be better to have a final column with this number. For example the CC 2D DARR at pH 7 is 40 scans x 860 rows (complex, presumably, so x 2) x 2.8 s = ~54 hrs (?). If so then this further shows that the 3Ds would be feasible.

Responses to reviewer comments

We thank the reviewers for their thoughtful comments and suggestions and we have revised the manuscript, taking all comments and concerns into account. Below we reproduce each reviewer comment, along with our detailed responses and a description of the changes that were made in the revised manuscript and supporting information.

Reviewer #1 (Remarks to the Author):

This is a great paper. The authors show quite convincingly, by ThT fluorescence, electron microscopy, and especially solid state NMR, that gamma-D-crystalline aggregates that form at physiological pH are definitely not amyloid-like. The data are clearly presented and solid, and interpreted appropriately. I almost never recommend in favor of publication without revisions, but in this case I must make an exception. This paper should be published in its current form.

We thank the reviewer for finding our work interesting and his/her overall positive response.

Reviewer #3 (Remarks to the Author):

The groups of van der Wel and Gronenborn present an interesting study of gammaD crystallin (GDC for this review) that potentially overturns a paradigm in the field of amyloids. The results are highly significant and the main point of the paper is hard to miss. Namely, at pH 7 the P23T mutant of GDC forms aggregates that look amorphous by microscopy but in fact have very high microscopic order as evidenced by the ssNMR spectra, but at lower pH the aggregates look more like typical amyloid based on the low resolution measures, yet have very poor microscopic order based on the ssNMR. The majority of the paper then is devoted to modeling these ideas and postulating on the mechanisms.

We appreciate that the reviewer acknowledges the importance of our key findings, agrees that they are “hard to miss”, and that they “potentially overturn[s] a paradigm in the field”.

Unfortunately, there are quite a few data sets and discussion points left out that one would need to make this study complete.

1. From the beginning, the structure of P23T GCD is discussed but not the wild type. Presumably there is a difference (albeit minor) in the structure that impacts the behavior of the mutant in forming aggregates.

This and several later comments express concern that our focus was strictly on the P23T mutant protein and the reviewer advocates for a more in-depth discussion of the aggregation behavior and molecular structure of the wild-type (WT) protein. These comments made us realize that we should provide more explanation why we are studying this mutant in its aggregated form, rather than WT. It is generally accepted that the highly soluble native WT HGD does not contribute to cataract formation. Indeed, there are data available that demonstrate that it is only when HGD protein's solubility is compromised by mutations or chemical changes that aggregation will occur.

Experimental studies show that WT HGD is one of the most soluble proteins known (several hundreds of mg/mL, which is orders of magnitude higher than the disease-associated P23T mutant protein). It even resists aggregation in the presence of aggregated mutant protein. Therefore, the aggregation by unmodified WT protein is not relevant to our understanding of cataract formation, and structural and biophysical studies focus on cataract-associated mutants, such as P23T.

In the revised manuscript we now describe the P23T mutant earlier in the Introduction, and we provide a more in-depth discussion of this variant's relationship to congenital cataract disease. These revised sections are on pages 3-4 of the Introduction.

At figure 1 it would be helpful to show the differences in the monomer structure, and later to include the WT chemical shifts in the comparison modeling.

We decided to not include any more analysis of the WT protein in our results and discussion sections, since this will distract from the main point of the paper and since we and others had several papers that dealt in detail with the WT protein. We have, however, revised the main text on top of page 4 to address the reviewer's concern.

“As examined in detail in prior work^{12,17-20}, this mutant protein is structurally very similar to the WT protein. This is reflected in a backbone RMSDs of 0.4 and 0.5 Å for the N-terminal (NTD; cyan) and C-terminal domains (CTD; pink), respectively (comparing the X-ray structures), or solution NMR per-residue chemical shift differences (Dd) of less than 0.2 ppm (except for the 0.4 ppm Dd for the residue after the P23T mutation)^{17,18}. Nonetheless, the mutant protein has lost the remarkable solubility of WT hgD and is associated with autosomal dominant congenital cataracts that form early in a child's development, affecting families across the globe^{12,17-21}.”

2. I am not sure the term “amorphous” should be used in the title and/or elsewhere since in fact this is a misnomer. Can the authors please address this point more explicitly somewhere?

One may indeed argue that the term “amorphous” for these aggregates is a misnomer. We now address this explicitly in the Discussion (on p. 17), where we state: “As such, at least in this case, the “amorphous” label appears to be a misnomer when it comes to the atomic level structure of the aggregated protein.”

Nonetheless, we do feel it is important to use this term, as it is widely-used terminology in the cataract/crystallin literature and also more broadly in the protein aggregation field. The word ‘amorphous’ is commonly used to describe a category of protein aggregates that lack a fibrillar appearance, as observed by EM. We have also expanded on this terminology in the Introduction section, where the term is first used (bottom of p. 4 and top of p. 5). In addition, we have changed the word “amorphous” to “amorphous-looking” in various places in the paper (including the title), to emphasize the fact that this term reflects the appearance, even if it does not reflect the internal structure.

3. A somewhat major concern is that the spectra are not assigned, although they could be. The 2D CC spectrum is very well resolved and many residue types give outlier CA/CB shifts; the CA-CO region is nicely separated and most of the Ala and Thr and Ser residues give clean peaks in the 2D. To perform now-standard NCACX and NCOCX spectra even as 2Ds would likely confirm the specific assignments. Much of the paper relies on comparison of modeled spectra with experimental spectra, and hypothesized assignments are indicated on several of the figures but it is not clear whether any of these signals were actually confirmed. If not, then (for example) in Fig 7 and S2 the labels should be removed or at least a caveat placed in the caption. Certainly if residues T23,

A63, I170 etc. have been identified with confidence, the logic should be elaborated upon somewhere. But the better solution would be to have the backbone correlations.

In order to alleviate any concerns by this reviewer, we now provide some assignments for selected resonances of the protein aggregate, based on added 2D and 3D heteronuclear ssNMR experiments. This new data is provided in Supplementary Table 2 and plotted in Supplementary Figures 4-5 in the SI. We note, however, that even with the remarkable spectral quality, complete assignments are non-trivial given the size of the protein (173 residues), and are beyond the scope of the present MS.

The assignments confirm prior assignments based on the 2D data (Supplementary Fig. 3), with two exceptions. Two sets of resonances exhibited low intensities, which made them hard to detect in the CC 2D spectrum. Aside from including these new assignments, we also revised Supplementary Figure 3 to clearly show these resonances. We also relabeled Fig. 7, to clearly differentiate between different assignment types by annotating them with distinct symbols. This is explained in the Figure caption and the text (p. 12; first paragraph).

4. Related to the point above, clearly the modeling gives a nice result but it is hard to assess the relative quality of the various models without quantitation. Methods are available for quantifying the agreement between models and experimental data of this type. It would strengthen the argument to provide such quantitation. I am sure the NMR experts see the agreement but maybe not so much the non-experts, especially since quite a few of the expected peaks are missing and/or slightly shifted. How would this compare with the WT shifts for example?

We added two new figures (Supplementary Figures 4 and 5) to the SI in which a more quantitative comparison between the two models is provided.

Our rationale for not including the comparison with the WT resonance frequencies was described above. Aside from the lack of disease relevance, we need to emphasize that the chemical shift differences between the WT and P23T HGD are much smaller than the differences between the solution and ssNMR chemical shifts.

5. Around lines 205-208 specifically the discussion is weak due to a lack of specific assignments. Resonance frequencies are changing, but for which residues? Again it is my view that these assignments can and should be made to strengthen the arguments here.

As detailed above, we now include assignment data obtained from 2D and 3D ssNMR experiments. This is included in the revised MS in the text and the figures (e.g. Fig. 7; Supplementary Figures 3-5; Supplementary Table 2). We would like to point out, however, that the new data did not change the conclusions of the MS.

6. Again around line 227-230 there is discussion that the ssNMR data “are unable to distinguish the different kinds of beta-sheet amyloid core architectures”. At minimum the authors should add qualifiers to this statement (“The ssNMR data we have obtained so far do not yet distinguish...”). But even so, the assignments alone would distinguish the cases propose on the basis of chemical shift perturbations relative to the solution data perhaps. Also a single long mixing time CC 2D spectrum would give cross peaks, very likely which would confirm or deny the postulated models. Mixed labeling would also inform upon this. So the authors could discuss these options and preferably in my view perform these experiments to test the hypothesis.

We have added the suggested qualifiers and the sentence in the revised MS now reads: “. The ssNMR data we have obtained thus far are unable to distinguish the different kinds of b-sheet amyloid core architectures.” on p. 13 (first paragraph).

The reviewer is correct that there are various ssNMR experiments that allow one to tell apart different amyloid architectures (some of which we have previously applied to a variety of different amyloids). It is clear, however, that unambiguous structure determination requires long-range distance measurements in fibrils, comprising mixed labeled proteins and/or PITHIRDS or DQ-DRAWS measurements with site-specifically labeled samples and multiple levels of isotopic dilution. We believe that this is not warranted, given that the biological relevance of the amyloid fibrils that form at pH 3 is unclear.

7. Again it is not clear (line 270) what is meant by chemical shift changes. Which surface-facing residues? Are these site-specifically assigned?

As noted above, we have added partial site-specific assignments that complement our existing analysis and thus enabled further quantitative analysis, which was not present in the original manuscript. The new data in the SI (Supplementary Figures 3 and 4) identify specifically those residues that exhibit different chemical shifts in solution and the solid state. The locations of the associated residues in the native fold are illustrated in the updated Fig. 7, and we comment on these data in the text of the revised MS (p. 12; final paragraph). They cluster on the outside surface of the NTD and CTD domains. Additional revisions of the text are in the Discussion, in both paragraphs of p. 14 and the top paragraph of p. 15.

8. Around line 276 there is stress on the hypothetical nature of the structure models. This should also be made plain on the figure caption and figure.

We have changed the figure caption of Figure 8 to clarify that the models reflect “proposed aggregation mechanisms.” We also added the words “hypothetical models” to the figure itself in the upper left-hand corner.

9. Line 292 mentions for the first time in this paper that the P23T “mutation itself does not cause much change in the structure”. First, what does “much change” mean here? Second, as noted already above

We now quantify the amount of change in the Introduction (p. 4; first paragraph), noting specifically the 0.4 and 0.5 Å backbone RMSDs for the NTD and CTD domains. See also our response to the second half of this reviewer’s Comment 1 (above).

10. I missed the experimental data for the WT protein. Was there any? Is the behavior at all unique to the P23T mutant? Would the same native fold aggregate be obtained from WT under these conditions? This seems to be a very important question to answer since the title implies unique behavior of this mutant. Exploring other mutants would be less essential, but a comparison with WT seems to be a fundamental part of the paper that is missing. (at least kinetic data such as in Fig S1)

We have already responded to this comment in our response to Comment #1 from this reviewer (above), noting that WT HGD is extremely soluble and that WT HGD does not aggregate to form cataracts. We also, expanded on the rationale why we study the disease-associated P23T mutant. The specific issue of WT solubility is now also addressed in the caption of Figure S1, as also pointed out in our reply to the next comment:

11. line 360: so if the concentration is increased up to the point that the solution becomes cloudy, without a defined end point of concentration based on the physiological case, then it would be expected that the WT would do the same thing but perhaps at a slightly higher concentration (according to Dobson all proteins do this eventually)

We appreciate the reviewer's concern, and we have expanded our discussion of the differences between WT and mutant HGD in the Introduction and throughout the MS (see Comment #1). The reviewer's assumption that the WT would aggregate at "slightly higher concentration" has not been observed (at least at physiological pH). We added text to that effect to the caption of Supplementary Figure 1 (which shows the aggregation assay results): "Note that the solubility of wild-type hγD is at least one to two orders of magnitude larger, with reported solubilities ranging from 170 to more than 400 mg/mL, depending on conditions^{3,4}." (p. S5 of the SI)

One well-known finding from the work of Dobson and colleagues suggests that under appropriately destabilizing conditions most (or many) proteins are capable of forming amyloid-like fibrils. We discuss this notion on p. 16 (second paragraph), including the requirement for destabilizing the native fold. This destabilization process is not active in this case, given that this comment is in reference to the aggregation behavior at neutral pH (in absence of denaturants).

More recently, Dobson and co-workers have discussed the notion of "supersaturation", arguing that proteins under physiological conditions (in vivo) are often present very close to their solubility point. In the case of HGD the physiological concentrations are in the hundreds of mg/mL (noted e.g. in the second paragraph of p. 3; bottom of p. 16), consistent with the abovementioned solubility levels of WT HGD that far exceed the solubility of the disease-associated P23T HGD mutant.

Minor points:

1. p. 7 line 111: "So-called" cross polarization?

We have removed the words "so-called" from this sentence.

2. p. 8 lines 148-149: Most Lys sidechains are mobile even if the backbone is not, so it is unclear that this is proof the NTD being highly mobile.

We re-examined the 2D INEPT/TOBSY spectrum (Fig. 5b) and ascertained that we observe backbone correlations for the Lys, indicating that not only the Lys side chain but also the backbone is mobile. Moreover, other residues' backbone carbon resonances are also visible in the spectrum, illustrating that these dynamics. We revised Fig. 5 and expanded the discussion of these results (p. 8 bottom; p. 9 top). This paragraph now reads:

"The spectra contain signals from both backbone and side chain atoms, including those of a Lys residue. Since it is the only Lys in the protein, those mobile Lys signals must stem from Lys-2 near the N-terminus of the NTD. Moreover, other visible peaks include Gly, Lys, Ile, Thr, and Leu residues, which happen to make up the initial five residues of the NTD (see Fig. 1a). Thus, these results are consistent with the initial segment of the NTD being unfolded and flexible, outside the mostly immobilized core of the amyloid fibril assemblies. Additional residues, such as Ser and Arg, are not found at the very N-terminus, indicating that other parts of the protein also end up in a similarly dynamic state."

3. Fig 7d has an incorrect label (L23, but it must be T23). Please check the others too.

We have corrected this error in the revised Fig. 7 and double checked the other labels.

4. line 337: Tewksbury? (or Andover?)

We have corrected this typographical error.

5. line 357: why not mix continuously?

The small sample volumes often prevent measuring the pH while a stirring bar is present and we believe that no significant errors are introduced by not mixing continuously.

6. Table S1 gives all the information needed to compute the total experimental time but it would be better to have a final column with this number. For example the CC 2D DARR at pH 7 is 40 scans x 860 rows (complex, presumably, so x 2) x 2.8 s = ~54 hrs (?). If so then this further shows that the 3Ds would be feasible.

We have added a column to Table S1, which includes this information.

We hope that the revised MS is now acceptable for publication and look forward to a positive response.

Reviewers' Comments:

Reviewer #3 (Remarks to the Author):

Thank you to the authors for a thorough response to all the inquiries raised in my previous review. I find the responses more than satisfactory and in particular the clarifications regarding the wild type protein behavior and the assignments made of the spectra add significantly to the clarity and impact of the manuscript. I think it is now ready for publication.